# Contribution of Genetic Polymorphisms in Human Health

**DOI:** 10.3390/ijerph20020912

**Published:** 2023-01-04

**Authors:** Pieranna Chiarella, Pasquale Capone, Renata Sisto

**Affiliations:** Department of Occupational and Environmental Medicine, Epidemiology and Hygiene, INAIL Research, Via Fontana Candida 1, Monteporzio Catone, 00078 Rome, Italy

**Keywords:** health, disease, lifestyle, genetic variability, environment, risk factor, precision medicine

## Abstract

Human health is influenced by various factors; these include genetic inheritance, behavioral lifestyle, socioeconomic and environmental conditions, and public access to care and therapies in case of illness, with the support of the national health system. All these factors represent the starting point for the prevention and promotion of a healthy lifestyle. However, it is not yet clear to what extent these factors may actually affect the health of an entire population. The exposures to environmental and occupational factors are several, most of which might be poorly known, contributing to influencing individual health. Personal habits, including diet, smoking, alcohol, and drug consumption, together with unhealthy behaviors, may inevitably lead people to the development of chronic diseases, contributing to increasing aging and decreasing life expectancy. In this article, we highlight the role of susceptibility biomarkers, i.e., the genetic polymorphisms of individuals of different ethnicities, with particular attention to the risk factors in the response to specific exposures of Europeans. Moreover, we discuss the role of precision medicine which is representing a new way of treating and preventing diseases, taking into account the genetic variability of the individual with each own clinical history and lifestyle.

## 1. Introduction

In the year 2020, over 4 million new cancer cases and 1.9 million cancer-related deaths were estimated in Europe. These data, found in the same year, have a key role to assess and monitor cancer-control measures across Europe [1]. In general, half of the overall cancer diagnoses have been identified as breast, colorectal, lung, and prostate, while the most cancer incidences were found in the female breast and the male prostate [2]. Collectively, all these four cancers account for half of the overall cancer burden in the European population, although in terms of death, the most common causes of cancer have been recognized in the lung, colorectal, breast, and pancreas. If we have a look at the life of older people in Europe, we see that the variability of life expectancy is not identical in both genders. This difference between men and women has the tendency to disappear after reaching 80 years of age. Furthermore, men and women seem to have different susceptibilities to disease, confirming that female life expectancy exceeds that of males in all European countries [3,4]. Apart from cancer, the main causes of death are circulatory diseases, followed by respiratory diseases [5]. Cardiovascular disease remains the most common cause of death worldwide and the most common cause of death in Europe (Figure 1). Previous studies have reported that cardiovascular disease kills nearly four million people in Europe every year, approximately 44% of all deaths, with ischemic heart disease accounting for 44% of these cardiovascular deaths and stroke accounting for 25% [6].

In elderly people, we also assist with several comorbidities, including neurological disorders such as senile dementia, Alzheimer’s, and Parkinson’s disease. Alzheimer’s disease accounts for 60% of all dementias. This pathology is considered a chronic degenerative disease in the general population and has been defined by the World Health Organization (WHO) as a deterioration in cognitive function [7]. At present, this morbidity has been considered a global public health priority. In 2015, 47 million people affected by a form of dementia were estimated in the world, over 1 million 200 thousand in Italy, with a prevalence in the over 65-year-old population of 4.4%. The prevalence of this pathology increases with age and is higher in women. In Italy, the prevalence in women ranges from 1.0% for the 65-69 age group to 30.8% for those over 90 years of age, compared to men, whose values vary from 1.6% to 22.1%, respectively, with about 900,000 people suffering from dementia, 600,000 of which with Alzheimer’s disease [8,9].

While it is easy to diagnose a specific disease in patients, it is much more difficult to identify a potential or ongoing disease in those people who are daily exposed to several unknown toxic and dangerous substances, in both environmental and occupational contexts. Although workplace exposure to chemical and physical agents is strictly controlled to guarantee the respect of specific Occupational Exposure Limits (OELs), it is fundamental to adopt the necessary measures to guarantee individual safety, including personal protective and collective equipment such as gloves, helmet, mask, and waterproof shoes and suits, which have been introduced to avoid the contamination with toxic substances as much as possible. It may be possible for a subject to be accidentally in contact with dangerous substances present indoors. However, the human body has many other defenses that depend on the presence of an efficient immune system, able to counteract any insult. Thanks to individual genetic inheritance, a subject will be able to repair potential damage by a variety of enzymes encoded by their own body, even though the effect of the exposure may be poorly known [10].

## 2. Causative and Susceptibility Genes

The human genome contains approximately three billion base pairs, which reside in the 23 pairs of chromosomes within the nucleus of our cells. Each chromosome contains hundreds to thousands of genes, which carry the instructions for making proteins. Each of the estimated 30,000 genes in the human genome forms an average of three proteins (National Human Genome Research Institute). Scientists have already discovered many functions of specific genes and the effects associated with variation in the human genetic code. It is well known that genetic variations underlie the great phenotypic diversity that we know well, such as eye and hair color, and not just physical traits. Therefore, there is no doubt that genes also contribute to modifying our character, personality, and vulnerability, considering also the influence of epigenetic factors, i.e., DNA methylation histone modifications and microRNA expression transmitted by parents to children.

The variability of genes in humans is widely known, as the differences in the phenotype among individuals are strictly related to the genotype, which is inherited from ancestors and depends also on specific ethnicities [11]. When the genetic basis of diseases is taken into consideration, it is possible to distinguish two types of genes: the causative and susceptibility genes. The causative genes are those that, if present in an altered form (i.e., a mutation), develop the associated pathology. This is the case, for example, of certain familial forms of Alzheimer’s dementia where only 5% of cases are linked to mutations of known genes, so the presence of the mutated allele is necessary and sufficient to develop the pathology condition [12]. In the case of the susceptibility genes, the presence of a defective “allele variant” of a gene does not mean the individual will necessarily develop the disease, but it will be more likely to develop it rather than in other individuals that do not have it. It is clear that other factors, such as genetics and environment, will contribute to causing some individuals to develop the disease while others do not. This is the case, for example, for the epsilon 4 allele of the apolipoprotein E (APOE) gene involved in fat metabolism where a subtype of the APOE gene is involved in Alzheimer’s disease and in cardiovascular diseases [13]. In fact, it was more frequent in patients with sporadic Alzheimer’s disease (in those patients who account for 95% of Alzheimer’s cases) but it is also present in healthy individuals who will never experience Alzheimer’s in their lifetime. In the case of Alzheimer’s, the presence of the mutated allele is neither a necessary nor sufficient condition for the manifestation of the disease [14]. In the last 30 years, gene polymorphisms have raised a lot of interest in many scientific fields related to both public health and disease. Gene polymorphisms are the most common type of genetic variation in humans. They are present in the human population at a frequency higher than 1% and differ from DNA mutations which are generally observed at extremely low frequencies and in a restricted number of individuals. Genetic polymorphisms are important contributors to interindividual variation since they have been investigated as useful biomarkers in the medical context as well as in the study of pathology, epidemiology, pharmacology, clinical immunology, and ethnicity. While gene mutation is rare and generally known to cause a genetic disease, gene polymorphisms are not necessarily associated with a specific disease [15].

## 3. Genetic Polymorphisms as Biomarkers of Susceptibility

Humans are exposed to a wide variety of environmental and occupational factors throughout their lifespan. These include both naturally occurring toxins and chemical toxicants like pesticides, herbicides, chemicals, and industrial products, most of which have been implicated as possible contributors to human disease susceptibility. In the case of the occupational setting, the dangerous substances are well known and manipulated with strict control, while according to the latest data, it has been estimated that about 24% of all diseases in the world are due to environmental factor exposure. Much of these risks could be avoided through targeted interventions, as confirmed by the World Health Organization report (WHO) entitled “Preventing Disease through healthy environments: towards an estimate of the environmental burden of disease” [16]. To give an example, the subjects could be exposed to a mixture of pesticides or a combination of neurotoxic chemical solvents used in several industries such as in transportation, mining, construction, manufacturing, and shipbuilding, whose applications vary from being used individually or in the form of a mixture, such as in glues, paints, and cleaning products. That said, gene polymorphisms have the power to identify susceptible subgroups in exposed populations, and if we know exactly the polymorphism function, it might be possible to identify a population at risk, due to the different allele frequencies among ethnic groups. However, if a single genetic trait can be associated with an increased risk in specific individuals or populations, these traits should be studied to evaluate the probability of contributing to the risk of developing a disease [17]. In our former study, we investigated the susceptibility risk in four ethnic populations, i.e., Africans, East Asians, Europeans, and South Asians, predicting a model to assess the susceptibility among such subgroups. In such context, we considered the most common genetic polymorphisms involved in the exposure of occupational settings. In particular, we analyzed the gene polymorphisms involved in the metabolism, i.e., (1) detoxification; (2) oxidative stress; (3) DNA damage’s repair [18]. Our previous findings reported different susceptibilities in the four ethnicities, demonstrating the highest relative risk related to the genes of detoxification, and oxidative stress was found in the South Asian population, and the highest risk, associated with the DNA repair gene, was instead observed in the Caucasian ethnic group [17,18]. The ethnicity to which the individual belongs contributes to the difference in the response to the exposure.

### 3.1. Genetic Polymorphisms and Differences in the Metabolism

There is a relationship between the genetic predisposition of an individual and their ability to metabolize a substance. Differences in drug metabolism can lead to severe toxicity or therapeutic failure due to a change in the ratio between the drug dose and the concentration of pharmacologically active substances in the blood as a result of genetic modifications [19]. Genetic polymorphisms of drug-metabolizing enzymes give rise to distinct subgroups in the population that differ in their ability to perform certain drug biotransformation reactions [20]. In general, five distinctive groups of metabolizers have been identified:(1)The extensive metabolizer (EM) typical of the normal population. These subjects are either homozygous or heterozygous for the wild-type allele and have a normal metabolism;(2)The slow metabolizer phenotype (SM) that is associated with the accumulation of specific drug substrates in the body, inherited as a recessive autosomal trait due to the mutation or deletion of both alleles showing a slow metabolism. In some patients, the drug is metabolized very slowly, accumulating the substance in the bloodstream;(3)The poor metabolizers (PM) carry two defective alleles, showing a complete absence of activity. The higher body concentration of the substance may cause adverse effects due to the substance accumulation;(4)The rapid metabolizers (RM) clear the drug very quickly, and the therapeutic concentration of the drug in the blood and tissues may not be reached. That means the subject should have a higher dose to produce an effect;(5)The ultra-extensive metabolizer (UEM) is characterized by enhanced drug metabolism due to gene amplification inherited as an autosomal dominant trait. Individuals with the ultra-extensive phenotype are prone to therapeutic failure because the drug concentrations in the plasma at normal doses are by far too low (faster metabolism) [21]. Here, we distinguish the most common behavioral habits in the following categories.

### 3.2. Metabolism of Drug

Drug metabolism describes the biotransformation of pharmaceutical substances in the body so that they can be eliminated more easily. The majority of metabolic processes involving drugs occur in the liver, as the enzymes that facilitate the reactions are concentrated there. The rate of drug metabolism can vary significantly for different patients. For instance, the CYP2D6 enzyme is responsible for the oxidative metabolism of 20–25% of drugs. The CYP2D6 iso-enzyme is by far the most extensively characterized enzyme from the CYP450 superfamily, which exhibits a polymorphic expression in humans. It accounts for not more than 2.6% of CYP450 in the liver and plays a very important role in the metabolism of almost a hundred of the most commonly used drugs [22].

### 3.3. Metabolism of Smoke

Tobacco consumption represents the main etiological factor in lung carcinogenesis and lung cancer is the most frequent malignant neoplasm in many countries. Other factors such as individual genetic susceptibility, environmental and occupational exposures, stressful life, poor diet, and many other factors may influence the quality of life of individuals. The European directive on the smoking ban was passed by the European Parliament and Council in 2003. The entry into effect of the EU’s Tobacco Advertising Directive occurred on 31 July 2005 (World Bank Report on the Economics of Tobacco Control, 1999) https://ec.europa.eu/commission/presscorner/dtail/en/IP_05_1013 (accessed on 2 September 2022) [23]. In the USA the main cancer-related cause of mortality worldwide in both genders accounts for an estimated 27% of total cancer deaths in 2015 and 20% in the EU in 2016 [24,25]. Nicotine is the primary psychoactive constituent of tobacco. Despite it is not a carcinogen, this substance is involved in smoking in addition to the continuous exposure to toxic agents present in tobacco smoke. Once inhaled, nicotine enters into the lungs by circulation to bind to nicotinic cholinergic receptors. The dominant pathway of nicotine metabolism in humans is the production of cotinine, which occurs in two steps. Cotinine is a nicotine metabolite used to quantify exposure to active smoke, and especially to passive smoking. The CYP2A6 enzyme is responsible for the majority of nicotine metabolism and is classified into CYP2A6 genotypes with predicted phenotype groups, as described for the CYP2D6 in the above paragraph [26]. Our analysis confirms that cancer risk due to smoking changes in different ethnicities. Among cigarette smokers, African Americans and Native Hawaiians are more susceptible to lung cancer than White people, Japanese Americans, and Latino people [27].

### 3.4. Metabolism of Ethanol

Dependence on alcohol may cause liver disease with a progressive inflammatory process. In particular, alcoholics may undergo hepatic steatosis, a reversible condition resulting in the accumulation of triglycerides in the liver. As a result, the individual may undergo an increase in hepatomegaly. Other negative effects resulting from alcoholism include cardiovascular disease, hypertension, lung inflammation, mood disorders, anxiety, depression, and memory loss [28]. The most relevant enzymes of alcohol metabolism are alcohol dehydrogenase (ADH) and aldehyde dehydrogenase (ALDH) with the contribution of cytochrome P450 (CYP2E1). In general, ethanol is metabolized by alcohol dehydrogenase (ADH) and by aldehyde dehydrogenase (ALDH) enzymes, where acetaldehyde is oxidized to acetate, while CYP2E1 metabolizes a small fraction of the ingested ethanol. The coding variants in both of these genes seem to be protective, decreasing alcoholism risk by increasing local acetaldehyde levels, either because ethanol is oxidized more rapidly or because acetaldehyde is oxidized more slowly. The balance between the rates of ethanol and acetaldehyde oxidation could be crucial in determining acetaldehyde concentrations within cells, such that small differences in the relative activities of ADH and ALDH might produce significant differences in acetaldehyde concentration [29]. The distribution of ADH1B and ALDH2 coding variants changes greatly among different populations; for both genes, the most common protective alleles are found in people of East Asian origin [30,31]. Variations in genes encoding other ADH enzymes influence alcoholism risk in other populations. For example, ADH4 variants strongly affect alcoholism risk in populations of European descent. There are also non-coding variants that may affect the risk of alcoholism [29]. Although variations in individual ADH and ALDH genes can affect the risk of alcoholism, we should think that one gene is not sufficient to determine the risk. Nonetheless, there are many genes unrelated to ethanol metabolism which may affect the risk of being influenced by multiple social and environmental factors. The level of ethanol consumption and the risk of alcoholism mainly depends on the ADH or ALDH alleles. ADH1B and ALDH2 have been reported as the genes most strongly associated with alcoholism risk. A variant of the ADH1B gene (rs1229984, i.e., Arg48His) has been reported to be associated with reduced rates of alcohol and drug dependence. The allele with increased activity and higher oxidation of ethanol to acetaldehyde is His48, encoded by rs1229984. Carriers with one or two ADH alleles, such as (G/A) or (A/A) have a reduced risk of alcoholism, metabolizing alcohol faster than carriers of the G/G genotype [32].

The most important information on the gene polymorphisms, relevant to the human metabolism, is summarized in Table 1 below. Here, we have focused on three categories of enzymes involved in the following functions: detoxification, oxidative stress, and DNA repair damage. Altogether, they include some functional enzymes involved in the metabolism of drugs, smoke, and ethanol, which affect the behavioral habits of many individuals. Such enzymes, summarized below, have been classified according to their specific function and allele frequency, showing specific effects depending on the amino acid substitution occurring in the polymorphism.

## 4. Discussion

The genome of each individual interacts with exposures to many environmental and occupational agents, including personal habits, such as diet, drug consumption, alcohol, and smoking. When a pathology by chance develops, all these factors influence several aspects of a complex response exerted by the human being: the age of onset, the rate of progression, and the therapies and side effects following medical treatments [62]. The actual knowledge of genetics that has been influenced by genome and environmental modifications will contribute to diagnosing and treating diseases in more precise and effective ways. The recent progress in medicine supported by novel technologies is developing fast, modifying the concepts and ideas to treat diseases. In the last thirty years, precision medicine started to emerge representing a new way to prevent aging and disease, taking into account many factors, i.e., the age, the genetic variability of the individual, the clinical history, the personal lifestyle and habit, and the effect of pharmacological treatments [63]. This novel concept is changing our perspective in terms of therapy and screening. If the traditional protocols for diagnostics and therapeutics have been generally structured on the basis of the average patient, precision medicine now shows a different perspective, in which the variability of the population and ethnicity include advancements in genome sequencing, allowing the discovery of novel mutations and other functional genes [64]. These polymorphic genes represent a group of susceptibility biomarkers for selected subgroups sharing common genetic characteristics [65]. The variability in the population represents a critical issue to develop targeted therapeutic protocols to treat diseases. In the study of disease, we need a global assessment that takes into account not only the clinical and instrumental examinations but also the patient’s history, familiarity, and lifestyle, together with all the factors affecting the progression of the disease and the response to treatment. Thanks to the identification of specific biomarkers, such as the biomarkers of exposure and effects in the occupational context, the risk of developing a certain disease can be earlier identified. This allows to improve preventive strategies or treatments that, if implemented early, will be more effective to treat disease supported by reliable diagnostic tests that will help to choose the best treatment. For instance, the analysis of the biological characteristics of a malignant tumor will allow to select patients who will benefit from personalized treatment. If this is not the case, the patient will be addressed immediately to alternative therapies. Reducing the side effects of a drug and adverse reactions is another goal. Pharmacogenetics tests may help to provide safe treatments with few adverse effects for the individual, taking into account the genetic profile. For instance, depending on the variant allele of polymorphism, the drug will be metabolized more or less quickly. This can explain why some patients will suffer from adverse effects, where the drug, poorly metabolized, is accumulating in the body, while other patients will metabolize very quickly in the absence of an effect. The reduction in the use of invasive tests in favor of safe molecular tests is another relevant issue. There are also pathologies, diagnoses, and follow-ups of dangerous tests, including biopsies, that are sometimes difficult to execute and too invasive for the patient. The use of reliable and harmless biomarkers of exposure and effect is mandatory in order to improve the quality of the patient’s life, choosing the most suitable biological matrix that should be easy to harvest, such as in the case of urinary sediment, exfoliated buccal cells and blood [66,67] matrices are not invasive, easy to harvest, facilitating the type of diagnosis. For instance, the use of oral mucosa exfoliates represents a valid alternative [68]. Differently from blood, the buccal cells are easy to harvest by self-made mouthwash or scraping, do not require specialized staff or equipment, and ensure good DNA yield with a low risk [66]. However, the most relevant advantage of using blood is the plasma collection, useful for biochemical analysis, and the harvesting of a high number of peripheral mononuclear cells that may be used for several DNA analyses, including genetic polymorphisms and molecular biology tests [69]. The only critical issue of blood accruing is the expertise of the operator who should avoid pain and the risk of infection for the patient. On the other hand, despite urine being rarely used for genotyping, due to the presence of a mixed cell population (leukocytes, renal tubular, transitional urothelial, and squamous cells), there are novel alternatives and useful kits to extract cells. However, no matter what biological matrix is used for genetic analysis, urine sample collection is mandatory for the assessment of internal dose biomarkers.

## 5. Conclusions

The effect of exposure to health hazards means that individuals and groups in the population are more or less likely to develop diseases after a potentially dangerous exposure. This condition, apparent or silent, is not only due to genetic inheritance but also to individual behavior and personal habits. People are more or less used to consuming drugs, drinking alcohol, and smoking [70]. In small amounts, the organism is able to detoxify the substance, but there are people with poorly efficient metabolisms due to their inherited genetic variability that differs from other individuals. Furthermore, the interethnic difference in the consumption of certain substances might have severe adverse effects or no efficacy at all, depending on the genotype. In the past, particularly around the year 1970, people did not pay particular attention to keeping themselves healthy [71,72]. Common habits and lifestyles changed many years later, when alcohol consumption, heavy smoking, and the abuse of psychotropic drugs were reduced [73]. However, despite the current perception of living with a correct habit and lifestyle, nobody can predict the duration of life expectancy. The last decade has seen many advances in proteomics and genomics science, and precision medicine emerged, representing a new concept of dealing with and preventing diseases, taking into account the genetic variability of the individual, the clinical history, and the lifestyle. Precision medicine is based on the decision to prescribe a specific drug suitable for the individual’s genotype with the aim of maximizing the efficacy of the drug by mitigating the risks of the ethnic group to which the patient belongs. In the absence of a genotyping test, ethnicity is seen as a model of the patient’s probable genotype, based on the frequency of genetic variations with some ethnic characteristics [68]. Precision medicine aims to develop effective strategies for patient groups who share specific genetic and molecular common traits. This novel science studies how the genetic structure of human beings influences the action of drugs administered to patients, with the ultimate goal of predicting and therefore preventing adverse reactions and/or therapeutic failures [74]. In particular, there is a need for the right drug against a specific disease with the correct dosage for patients who share common characteristics. The idea of precision medicine will change the approach to the prevention, diagnosis, and treatment of diseases. This method involves a significant change of perspective: traditional diagnostic and therapeutic protocols have been generally structured on the basis of the “average patient”, while precision medicine intends to take into account the variability of the population, in order to develop targeted therapies for selected subgroups with common traits. The genetic variability of a population, including the polymorphic genes presented in this article, becomes fundamental in order to develop tailored therapeutic protocols. Anyway, the right drug should be available at the right dose and to the right genotype. Recognizing the importance of interethnic differences in drug response, it is not surprising that regulatory authorities will require the adequate participation of various demographic subgroups of patients by gender, ethnicity, and age in clinical trials. This is fundamental to assessing safety and efficacy data in these subgroups [75,76].

## Figures and Tables

**Figure 1 ijerph-20-00912-f001:**
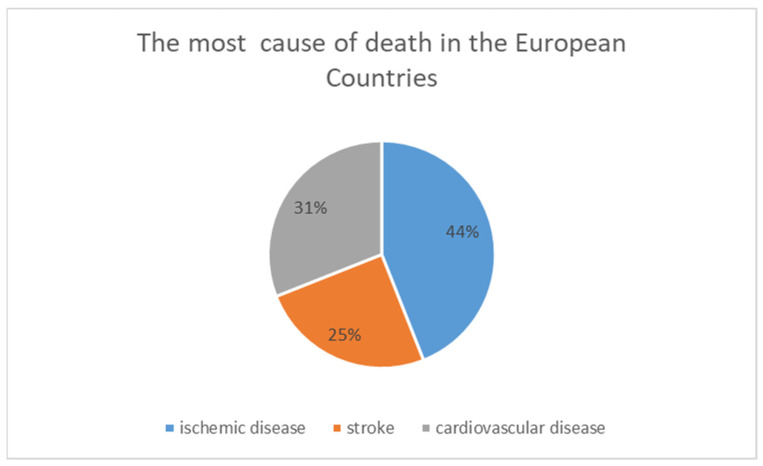
Distribution of death for cardiovascular, ischemic, stroke, diseases.

**Table 1 ijerph-20-00912-t001:** List of specific gene polymorphisms and relative enzymes relevant to the human metabolism.

Polymorphism Function	Detoxification	European Allele Frequency	Effects on Enzyme Activity
**Glutathione S Transferase**			
**GST-T1** rs17856199	Detoxification of xenobiotics, carcinogenic substances, therapeutic drugs, environmental toxins, and oxidative stress products.	GST-T1 A = 0.867 C = 0.183	GST-M1 and GST-T1 homozygous deletions (pos/null) have a decreased ability to detoxify carcinogens, toxicants, and oxidative stress products. The gene deletion has been found in Caucasian and Asian populations compared to Africans [33].
**GST-M1** rs366631		GST-M1 A = 0.488 G = 0.512	
**GST-A1** rs3957357		GST-A1 A = 0.571 G = 0.429	The distributions of GSTA1-69C > T promoter haplotypes and diplotypes were significantly different among the human populations. The frequencies of GSTA1-69C > T polymorphism were similar to those of the African American population and the populations with White ancestry, but significantly different from those reported for the populations with Asian ancestry [34].
**GST-P1** rs1695		GST-P1 A = 0.669 G = 0.331	Several papers published findings associating GSTP1 Ile105Val genotypes with bronchial, childhood, or atopic asthma. Compared with the AA genotype, the GA + GG genotype decreased lung cancer susceptibility [35].
**Epoxide hydrolase**			
**EPHX1-Ex_3** rs1051740	Biotransformation enzymes converting epoxides from the degradation of aromatic compounds to trans-dihydrodiols excreted from the body. EPHX1 was shown to take part in protection against oxidative stress.	EPHX1-Ex3: T = 0.696 C = 0.304	In vitro polymorphisms in exons 3 (His113Tyr) and 4 (Arg139His) lead to reduced activity (slow allele) and increased activity (fast allele). T to C substitution (slow allele) reduces the enzyme activity. Decreased activity (histidine) A to G substitution (fast allele) increases the enzyme activity. Increased activity (arginine) [36].
**EPHX1-Ex_4** rs2234922		EPHX1-Ex4: A = 0.836 G = 0.164	
**Cytochrome P450 family**			
**CYP1A1_2A** rs4646903	Catalysis of reactions involved in the drug metabolism and synthesis of cholesterol, steroids, and lipids, some of which are found in cigarette smoke. The enzyme’s endogenous substrate is able to metabolize some polycyclic aromatic hydrocarbons into carcinogenic intermediates.	CYP1A1-2A: A = 0.893 G = 0.107	The CYP1A1 Ile462Val polymorphism may enhance the susceptibility to cervical cancer in Caucasian females. The gene has been associated with lung cancer risk. Higher inducibility. Increased oxidation [37].
**CYP1A1_2C** rs1048943 Ile462Val		CYP1A1-2C: T = 0.965 C = 0.035	
**Cytochrome P450 2E1**			
**CYP2E1*5B** rs3813867	Cytochrome P450 2E1 (CYP2E1) is one of the major enzymes involved in the metabolism and detoxification of various drugs and xenobiotics.	CYP2E1*5B: G = 0.959 A = 0.041	The genotype distributions of CYP2E1*5B and *6 were similar to the Caucasian population but were different from East Asian populations. Wang, L.N.; Wang, F.; Liu, J.; Ying-Hui, J.; Fang, C.; Ren, X.Q. CYP1A1 Ile462Val polymorphism is associated with cervical cancer risk in Caucasians not Asians: A Meta-Analysis. Front. Physiol. 2017, 8, 1081. https://doi.org/10.3389/fphys.2017.01081. [38]
**CYP2E1*6** rs6413432		CYP2E1*6: T = 0.908 A = 0.091	CYP2E1*6 polymorphism causes a reduction in CYP2E1 enzyme activity [39].
**Cytochrome P450 family 2 subfamily D6 and Cytochrome P450 family 2 subfamily A6**			
**CYP2D6** rs16947	CYP2D6 is responsible for the oxidative metabolism of 20–25% of drugs.	CYP2D6: G = 0.710 A = 0.290	Unfunctional alleles represent 26% of the variability mainly in CYP2D6*4. “Frequency of CYP2D6 Alleles Including Structural Variants in the United States” Del Tredici, A.L.; Malhotra, A.; Dedek, M.; Espin, F.; Roach, D.; dan Zhu, Guang.; Voland, J.; Moreno, T.;A. Front. Pharmacol., 2018 Pharmacogenetics and Pharmacogenomics [40]
**CYP2A6** rs1801272	Nicotine metabolism is mediated primarily by cytochrome CYP2A6. The genetic variation in this gene has been linked with several smoking behavior phenotypes.	CYP2A6: A = 0.973 T = 0.026	The frequencies of this allele vary considerably among different ethnic populations, the deletion alleles being most common in Oriental people (up to 20%). Studies of Japanese populations suggest that CYP2A6 poor metabolizer genotypes result in altered nicotine kinetics and may lower cigarette smoking-elicited lung cancer risk, whereas similar studies in Caucasian populations have not revealed any clear associations between variant CYP2A6 genotypes and smoking behavior or lung cancer predisposition [41].
**N-Acetyl Transferase**			
**NAT1** rs4987076	N-acetyltransferase 1 detoxifies many drugs and chemicals found in the environment eliminated from the body or bioactivated to metabolites causing toxicity/cancer. NAT1 activity is regulated by genetic polymorphisms as well as environmental factors such as substrate-dependent down-regulation and oxidative stress.	NAT-1: G = 0.974 A = 0.025	NAT-1: a monomorphic form of the enzyme [42].
**NAT2** rs1208	N-acetyltransferase 2 enzyme detoxifies xenobiotics (carcinogens and drugs). Variation at the NAT2 gene has been linked to the human acetylation capacity: slow, intermediate, and fast, which modifies susceptibility to cancer and adverse drug reactions.	NAT-2: G = 0.434 A = 0.565	Multiple NAT2 alleles (NAT2*5,*6, *7, and *14) have substantially decreased acetylation activity and are common in Caucasian people and populations of African descent. Link between acetylator phenotype and increased risk for bladder and colon cancer [43].
**Oxidative Stress**			
**MPO** Myeloperoxidase rs2333227	Oxidoreductase catalyzing H_2_O_2_ to H_2_O.	MPO: C = 0.910 T = 0.089	A study of 127 Finnish patients concluded that the rs2333227 allele increased the risk of Alzheimer’s disease [44]. In a further study polymorphism of MPO was statistically associated with AD in a gender-specific manner. Myeloperoxidase polymorphism is associated with a gender-specific risk for Alzheimer’s disease [45].
**Heme Oxygenase** rs2071746 -413AT	Cytoprotective enzyme activated during cellular stress such as inflammation, ischemia, hypoxia, hyperoxia, and radiation.	HO1: A = 0.440 G = 0.560	In the Ala16Val, the Ala amino acid seems to be more favorable than the Val amino acid. Higher risk for Alzheimer’s disease (TT) [46].
**SOD2** rs4880	Mitochondrial enzyme with a key role in protecting the cell from oxidative damage.	SOD: A = 0.502 G = 0.498	Ala16Val: The Val amino acid is less favorable. Association of smoking and homozygosity for the MnSOD Val allele contributing to an increased risk of diabetic nephropathy [47].
**NRF2** rs6721961	Regulator of the cell transcriptional response to oxidative stress induced by exposure to xenobiotics.	NRF2 rs6721961 G = 0.875 (wt) T = 0.125 (mut)	The “G” and “T” alleles resulted in higher and lower expressions of NRF2 mRNA, showing that the G allele is beneficial for protection from pathologies. In contrast, the “T” allele of rs6721961 significantly increases susceptibility to the elevation of the hearing threshold at 4 kHz in the occupational setting [48]. NRF2 (rs6721961) (-617A/A) alleles in the NRF2 gene have been associated with female non-smokers with adenocarcinoma and are regarded as a prognostic biomarker for assessing the overall survival of patients with lung adenocarcinoma [49]. “C” indicates the wild allele, and “T” indicates the mutant allele [50].
**NRF2** rs6706649		NRF2 rs6706649 C = 0.882 T = 0.118	
**NRF2** rs35652124		NRF2 rs35652124 C = 0.327 T = 0.672	
**NQO1** rs1800566	Enzyme involved in preventing free-redox radical generation.	NQO1 rs1800566: G = 0.809 A = 0.190	The variant enzyme, C609T, is ubiquitinated by the proteasome (greater risk for 609TT). Susceptibility risk for hepatocellular carcinoma [51].
**NQO1** rs1131341	(C465T, Arg139Trp) is a polymorphism within NQO1 (NAD(P)H dehydrogenase (quinone 1)	NQO1 rs1131341 G = 0.961 A = 0.038	The variant enzyme C465T shows reduced enzyme activity. Potential risk of (ALL) Acute Lymphoblastic Leukemia (AML) Acute Myeloid Leukemia [52].
**DNA damage Repair**			
**hOGG1** rs1052133	The human 8-oxoG DNA glycosylase 1 plays a central role in repairing 8-oxoGs via the base excision repair pathway. Evidence suggests that hOGG1 affects the activity as a genetic marker for the prediction of personal susceptibility to several cancers.	hOGG1 C = 0.778 G = 0.221	In solid tumors, the genotypes CG and/or GG at hOGG1 rs1052133 have been reported to be associated with an increased risk of various types of cancer. The polymorphic site of hOGG1, i.e., rs1052133 (Ser326Cys), C to G showed that the glycosylase activity of the “G” variant is more sensitive to inactivation by oxidizing agents than that of the “C” wild-type, and cells carrying the “G” allele may accumulate mutations more readily under oxidative stress [53].
**XRCC1** rs25487 Arg399Gln	The protein encoded by XRCC1 is involved in the repair of DNA single-strand breaks formed by exposure to ionizing radiation and alkylating agents.	XRCC1 C = 0.642 T = 0.357	XRCC1 has protective gene polymorphisms due to the enhanced repair activity even though the Arg399Gln polymorphism (rs25487 and rs1799782 Arg194Trp) have been reported to be related to prostate cancer [54].
rs1799782 Arg194Trp		G = 0.937 A = 0.062	
**XRCC3** rs861539	The gene is involved in the homologous recombination repair pathway (HRR) of double-stranded DNA, deputed to repair chromosomal fragmentation, translocations, and deletions.	XRCC3 G = 0.617 A = 0.382	XRCC3 promotes the homology-directed repair of DNA damage in mammalian cells. Three amino acid substitution variants of DNA repair genes (XRCC1 Arg194Trp, XRCC1 Arg399Gln, and XRCC3 Thr241Met) showed an association with breast cancer susceptibility. Polymorphisms of XRCC1 and XRCC3 genes and susceptibility to breast cancer [55].
**XPD/ERCC2** rs13181	The ERCC2 contributes to the synthesis of XPD protein. This is an essential subunit of a group of proteins known as the TFIIH complex with two major functions: the activation of gene transcription and the repair of damaged DNA.	XPD/ERCC2 T = 0.626 G = 0.373	XPD protein is involved in transcription initiation and in the control of the cell cycle and apoptosis. The ERCC2-rs13181 C allele was associated with a significantly increased risk of colorectal cancer risk [56].
**Alcohol dehydrogenase**			
**ADH1B** rs1229984 Arg 48 His	Enzymes involved in alcohol metabolism. The functional variant, Arg48His (rs1229984), is located in the ADH1B gene and protects against alcohol dependence.	ADH1B G = 0.951 A = 0.048	ADH1B has been reported to associate with reduced rates of alcohol and drug dependence. The allele with increased activity, meaning the more rapid oxidation of ethanol to acetaldehyde, is His48, encoded by rs1229984. Carriers with one or two ADH alleles, such as (G/A) or (A/A) have a reduced risk for alcoholism. These findings support that the His allele can greatly lower the risk of AD and alcohol abuse and provide strong evidence for the involvement of the ADH1B gene in the pathogenesis of AD and alcohol-induced diseases in particular in multiple populations such as Asian populations [57].
**ADH1B** rs2066702 Arg370Cys		ADH1B G = 0.997 A = 0.002	ADH1B Arg370Cys is monomorphic in European populations, but it has been shown to have a small effect on the rate of alcohol elimination in African Americans [58].
**ADH1C** rs698 Ile350Val	The rs698(A) allele is the most common, encoding isoleucine; the rare (T) allele encodes the variant phenylalanine	ADH1C A = 0.609 T = 0.391	An increasing number of studies have investigated the association between ADH polymorphisms and cancer risk in humans. AA normal enzyme activity AT-TT reduction in the enzyme activity and intolerance to ethanol. Among them, studies of the ADH1C Ile350Val (rs698) variant accounted for more than others. The results indicate that ADH1C Ile350Val polymorphism may contribute to cancer risk among African populations and Asian populations [59].
**ADH4** rs3805322	The alcohol dehydrogenase (ADH) family represents one of the key sets of enzymes responsible for the oxidation of alcohol. This enzyme is an important member of this family, it is a functional candidate for alcohol dependence in human consumption of alcohol.	ADH4 A = 0.998 G = 0.003	A significantly increased risk of developing esophageal squamous-cell carcinoma was revealed in subjects with the AA genotype for rs3805322 (ADH4) compared with those with the AG or GG genotype [60].
**Aldehyde dehydrogenase**			
**ALDH2** rs671 Glu504Lys	Aldehyde dehydrogenases efficiently oxidize and, in most instances, detoxifies a significant number of chemical aldehydes which otherwise would be harmful to the organism.	ALDH G = 1.000	Individuals with the ALDH2 504Lys variant were less associated with alcoholic liver disease compared to those with ALDH2 504Glu by genotypic and allelic analyses [61].

## Data Availability

Genotype and allele frequencies have been down-loaded from the Ensembl project of genome databases for vertebrates and other eukaryotic species http://grch37.ensembl.org/Homo_sapiens/Variation, public database (accessed on 2 September 2022).

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
