# Peer review of "Contribution of Genetic Polymorphisms in Human Health"

_ijerph, 2023, doi:10.3390/ijerph20020912_

Round 1

Reviewer 1 Report

Chiarella et al. present a review that highlights the role of susceptibility biomarkers in response to specific exposures for Europeans. Addressing the following points can help sharpen the review:

1.     In line 36, it should be 80 years of age.

2.     In line 54 it should be 600,000.

3.     It will be beneficial to summarize the percentage occur of various diseases in Europe mentioned in the Introduction paragraph in a pie chart.

4.     Line 67, 89, 95, 104, 147, 198, 220, 229, 232, 322 need a reference.

5.     Line 79 needs to be elaborated further to explain what the authors mean by character and personality.

6.     In line 100, it should be Alzheimer’s.

7.     In line 101, period should come after the reference.

8.     Line 113-123 need to be rewritten for clarity. It is hard to follow this part in it’s current form.

9.     In line 144, it should be “their” instead of “his” for gender neutrality.

10.  Check parentheses in line 188 and 189.

11.  In section 3.2, please add a section about the role of ethnicity in drug metabolism.

12.  In the introduction, the authors mention about the role of gender and age in diseases, however there is no mention of how they impact genetic polymorphisms in the review.

13.  The title should be modified to reflect the content of the review. The review focuses on gene polymorphisms associated with metabolism, but the title is very broad.

14.  In Table 1, left most column should be labelled as gene polymorphism and not detoxification.

15.  In table 1, format allele frequency table to be consistent to have one base per row.

16.  Italicize words in vitro in Table 1.

17.  Expand PAH in Table 1.

18.  In Table 1, add a column for indicating the diseases and/or impact of the change in enzyme activity and converting the enzyme activity column to bullet points can help increase the clarity of the content.

19.   Lines 306 to 312 are vague and need citation and clarity.

20.  Grammatical errors, typos and punctuations should be thoroughly checked for in the entire manuscript.

Reviewer 2 Report

The comments are in the document attached.

Reviewer 3 Report

Dear Authors, Thanks for the informative paper

There are some issues;

You mentioned European population in title of paper.

-        but I could not receive any important point in discussion or conclusion according to your title. Is there any difference between the European population and other populations? What's it? How does it influence the association between different risk factors and diseases in this population?

-        Maybe you could add a section on studies related to European populations (suggestion).

Round 2

Reviewer 1 Report

The authors have addressed several comments regarding the manuscript. However, they are yet to address some comments that were already mentioned previously. The manuscript is close to publication and needs these points to be addressed:

1.     In line 34-35, it should be 80 years of age.

2.     In line 58, it should be 600,000.

3.     In Line 82, add the description of character and personality to the manuscript or provide citation.

4.     In line 103, it should be “Alzheimer’s”.

5.     In line 118, it should be “most of them”.

6.     In line 123, no comma is required after risks.

7.     In line 144, it should be “their” instead of “his” for gender neutrality.

8.     Italicize words in vitro in Table 1 in row EPHX1-Ex_3.

9.     Expand PAH in Table 1 in the manuscript.

10.  Line 316 needs citation.

11.  Grammatical errors, typos and punctuations should be thoroughly checked for in the entire manuscript.

Author Response

Dear Reviewer 1, thank you for the comments of our manuscript. We think the paper  has now been improved. In order to find all the suggested corrections, we highlighted  on the manuscript all your requested points in red colour for your convenience.  Two further References have been included (Ref. 71 and 72)  Please see the attached manuscript. If the paper will be accepted we will keep in touch with the IJERPH journal to modify the corrections made from red into black colour.

Thank you again, Best Regards. 

Reviewer 3 Report

This version is more acceptable, thank you, dear authors.

Author Response

The manuscript has been modifies according to the requests of reviewer n. 1. All the modifications are highlighted in red colour for your convenience. Thanks for the suggestion to improve our review article. We specify that the number of reference is 76. It has been a pleasure to collaborate with the reviewer and with IJERPH journal. Best regards.
